# Patient-Reported Outcome Measures (PROMs) for Two Implant Placement Techniques in Sinus Region (Bone Graft versus Computer-Aided Implant Surgery): A Randomized Prospective Trial

**DOI:** 10.3390/ijerph17092990

**Published:** 2020-04-25

**Authors:** Ghazwan Almahrous, Sandra David-Tchouda, Aboubacar Sissoko, Nathalie Rancon, Jean-Luc Bosson, Thomas Fortin

**Affiliations:** 1Department of Oral Surgery, Dental School, University Claude Bernard, 69003 Lyon, France; dr.g.mahrous@gmail.com; 2ThEMAS TIMC UMR CNRS 5525, Grenoble Joseph Fourier University, 38041 Grenoble, France; sdavidtchouda@chu-grenoble.fr (S.D.-T.); jean-luc.bosson@imag.fr (J.-L.B.); 3Medical-Economic Evaluation Unit, University Hospital of Grenoble, 38700 Grenoble, France; 4Cellule Data Stat, University Hospital of Grenoble, 38700 Grenoble, France; asissoko@chu-grenoble.fr; 5Department of Oral Surgery, Hospices Civils, 69003 Lyon, France; nathalierancon@orange.fr

**Keywords:** dental implants, surgical guide, 3D imaging, planning software, sinus graft

## Abstract

Purpose: To assess patient-reported outcomes measures (PROMs) for two implant placement techniques in cases of sinus bone atrophy (bone graft surgery (BGS) versus computer-aided implant surgery (CAIS)), after surgery and one year later, and to evaluate the clinical success of both treatments. Methods: Sixty patients with bone atrophy in the posterior maxilla and in need of implant placement were randomly assigned to two groups, and in accordance with the case report form (CRF), 30 were treated with BGS and 30 with CAIS. Immediately after treatment and one year later, PROMs were assessed, and the clinical success of both treatments was evaluated. Results: No significant differences were found between BGS and CAIS with regard to the following: loss of implants (*p* = 492); patient recommendation (*p* = 210); duration of surgery (*p* = 987); pain on the intervention day (*p* = 512); pain in the week after intervention (*p* = 299); and complications in the stage of surgery (*p* = 1.00). Similarly, at one year, no differences were found with regard to the following: pain around implant (*p* = 481); infection of implants (*p* = 491); abnormal radiographic imaging (*p* = 226); occurrence of undesirable events (*p* = 1.00); loss of one of the implants (*p* = 1.00); plaque detection (*p* = 1.00); bleeding on probing (*p* = 236); and presence of keratinized mucosa (*p* = 226). However, a significant difference was found among BGS and CAIS with regard to the number of consultations (*p* = 0001); number of implants placed (*p* = 033); and treatment difficulty (*p* = 0369). Significant differences were found for peri-implantitis (*p* = 0481) and radiology of craterization (*p* = 020) in clinical examination at the first year. Conclusion: Treatment difficulty and number of consultations were higher for BGS than for CAIS, as well as peri-implantitis and bone craterization at one year, indicating significant differences between the two treatments. However, there were no statistically significant differences between BGS and CAIS regarding the other PROMs, at placement and after one year.

## 1. Introduction

The digital revolution has deeply transformed the world of dentistry. The introduction of novel devices such as intraoral [1], desktop [2], and face scanners [3]; computer-assisted design/computer-assisted manufacturing (CAD/CAM) software [4]; milling units [5]; and 3D printers [6], together with new materials [7,8], has changed the face of dentistry.

Cone beam computed tomography (CBCT) also has a deep impact in the implant dentistry because it allows the clinician to capture 3D information of the available bone for a better treatment planning, with considerably reduced radiations to the patient [9]; CBCT opens the door to computer-aided implant surgery (CAIS), which is nowadays one of the most important applications of digital dentistry [10].

Since of the first publication on image-guided surgery [11], many applications have been proposed to planimplant placement in advance. The analysis of aspects such as patient convenience, surgical and/or prosthetic complications, time efficiency, and costs, usually called patient-reported outcomes measures (PROMs), was carried out and few studies investigating static CAIS for PROMs were found [12].

Sinus atrophy is an anatomical obstacle and prevents the placement of implants of adequate length. In addition, the alveolar bone height and bone density in the posterior maxilla often lead to reduced implant-success rates. Therefore, it is of great importance that this obstacle be removed using adequate surgical procedures, with these procedures aimed at reducing the expanded volume of this cavity either partially or totally [13].

Several treatment options have been used in the posterior maxilla to overcome the problem of insufficient bone quantity. Conservative treatment has developed short implants to avoid entering the sinus cavity. Grafting the floor of the maxillary sinus has become the most common surgical intervention for increasing alveolar bone height prior the implant placement of adequate length. Sinus lifting procedures are performed routinely to provide the required height of proper and stable bone tissue around inserted dental implants [14,15], and the maxillary sinus augmentation/elevation graft is a procedure that greatly benefits patients, with a predictable outcome [15].

The ABC sinus augmentation classification is one of the classifications used to treat the posterior maxilla with factors critical for implant success. Class A: represents abundant bone with 10 mm bone height below the sinus floor and 5 mm bone width, allowing proper implant placement. Class B: indicates sufficient bone with 6 to 9 mm bone height below the sinus floor, and this can be further subclassified into division h (horizontal defect; 5 mm bone width), division v (vertical defect; 3 mm away from cementoenamel junction), and division c (combined horizontal and vertical defect). Class C: indicates compromised bone with 5 mm bone height below the sinus floor, and this can also be subclassified similar to Class B. The ABC classification is a simple system to guide clinicians in proper implant treatment of the posterior maxilla [16].

The use of an image-guided system for oral implant placement in the atrophic posterior maxilla reduces the duration of surgery and treatment by eliminating the graft healing period, patient and practitioner discomfort, and risks of morbidity. One of the main advantages of computer-guided technology in implant dentistry is the better control of the implant axis in relation to the prosthetic tooth position. This leads to a higher predictability of the treatment outcome with subsequent better patient information about the aesthetic final result [17].

Methodological reviews were recently focused on the accuracy and precision of computer technology applications in surgical implant dentistry with an overall mean accuracy of about 1.09 mm at the entry point and 1.51 mm at the implant apex for static computer-assisted implant placement [18,19]. Recent systematic review identified pain as beneficial advantages for image-guided surgery with flapless approach. Nevertheless, for the economic aspect, cost, and time efficiency, the literature is scare [12,20]. However, this is a key factor for any technique because, without adequate funding, it cannot be utilized, regardless of the benefits.

The aim of this prospective, single-center, randomized, therapeutic evaluation for patient-reported outcome measures (PROMs) is to compare the characteristics of surgical interventions for two implant placement techniques in the case of atrophied posterior maxilla. The first technique implied grafting the floor of the maxillary sinus to increase the alveolar bone and the other used 3D software planning and computer-guided template-based techniques. Statistically, the null-hypothesis determined that there were no significant differences between the two treatment methods and the technique of implant placement did not have an effect on the characteristics of surgical intervention.

## 2. Methods

### 2.1. Study Design, Inclusion/Exclusion Criteria

Sixty patients had edentulous posterior maxilla and came to the Center for Dental Care, Teaching, and Research, University Claude Bernard (Lyon, France) in order to undergo implant placement. All of the patients wanted a treatment using dental implants and had a strong bone resorption in sinus region and all of them were categorized C according to the ABC classification (the height of the alveolar bone was less than 5 mm). This was a randomized clinical trial in the Center for Dental Care, Teaching, and Research, University Claude Bernard Lyon 1 and was supported by the Civil Hospices of Lyon, France, under the name SINIMAGE (REF HCL: 2008.514/15). The study was approved by the local ethical committee (CPP 08/095, Ref. A 08-230, 9 December 2008) and followed the principles of the Helsinki Declaration. All patients were fully informed about the study and provided written informed consent. A random draw with sealed envelopes assigned the treatment of the patients. Two treatment techniques were allocated within the SINIMAGE study, the first by sinus lift and bone graft surgery (BGS) and the second by computer-aided implant surgery (CAIS). All of the patients were eligible for the inclusion and exclusion criteria of the study (two years was the period of acceptance of inclusion within the study). The first patient joined study in 07/2009 and the last patient’s final dental prosthesis was loaded in 2015. Annual clinical examinations after prosthetic implant placement were performed, and according to the SINIMAGE protocol, the annual examinations were conducted over the following five years (Figure 1).

The inclusion criteria were as follows:Patients over 18 years of age;Patients with an indication of sinus bone graft;The scanner examination should show that the patient could be treated with CAIS;Patients with free posterior maxillary, without extraction within the last three months;Patients with occlusion allowing the non-contact prosthesis in lateral movements;Antagonistic arcade natural teeth or implants;Had stopped tobacco for at least three months.

The exclusion criteria were as follows:Patients unable to understand the information given by the doctor for legal, psychological, and linguistic reasons;Difficulty of follow-up (impossibility or insufficient motivation);Pregnancy;Patients at risk of infective endocarditis, transmission of Creutzfeldt–Jakob disease;Patients with severe or acquired immunodeficiency;Patients with malignant disease, history of radiotherapy in the mandible region;Patients with severe hemopathy, hemophilia, chronic renal failure, autoimmune disease, a disease that had required organ transplant, poorly controlled diabetes, osteoporosis, rheumatic arthritis, or psychiatric illness;Patients under antimitotic or immunosuppressive therapy, under high doses of corticosteroids;Drug-addicted patients;Tobacco users smoking within the three-month restriction period;Imprisoned persons.

### 2.2. Surgical Procedures

All of the patients needed the implant placement and it could be performed using two techniques, BGS or CAIS. The reading of panoramic images and the planning software confirmed this need. Then, after identifying the patients, a random drawing using sealed envelopes divided patients into two treatment groups: CAIS or graft.

The first group (BGS) was based on 30 patients, who were treated using sinus lifting procedures through bone graft. A two-stage technique was followed. First stage: the bone graft was performed for lifting the Schneiderian membrane and then rigidly secured. The dental surgeon treated lift sinus with the creation of a window in the lateral sinus wall and placed the bone graft under the sinus membrane, and then tightly closed the surgery flap. Second stage: bone integration normally occurs 6 to 8 months after grafting and the implant placement is done after bone integration. A sinus scan should be taken before the second stage. After anesthesia was obtained, the dental surgeon drilled the pilot hole in the maxillary crest, and then completed the drilling position to reach the suitable diameter in order to insert the implant. Finally, a healing screw was placed to tightly close the surgery flap. Augmentation of the maxillary sinus floor is a well-documented technique and is generally accepted as a pure implantology procedure to facilitate placement of dental implants in the posterior atrophic maxilla [21].

The second group (CAIS): 30 patients were treated by computer-aided implant surgery (CAIS) technology (3D planning software, 3D imaging by cone beam computed tomography (CBCT), guided surgical template, and computer-aided surgery). Depending on the 3D images of patients by CBCT, surgical guides were created. EasyGuide protocol (Keystone-Dental, Burlington, MA, USA) was used. Prior to surgery, the template with resin cube (fiducially marker) was used in the 3D scanning and the EasyGuide planning software was used to transfer the planned implant placement to the surgical site. The template was drilled using a digitally controlled drilling machine in compliance with planned implant placement and the drilling machine made holes on the template and on the plaster cast. After anesthesia was obtained, the drilled template was placed and fitted in the mouth in the same position as during the template scan. The dental surgeon inserted the drill sleeves into the template holes and the drill was passed through them to create a cavity at the top of the crest. This small cavity acted as the entrance for the pilot holes after removal. The implants positions were then prepared using the pilot hole and the implants were inserted. Finally, a healing screw was placed and tightly closed at the surgery flap. The use of computer-assisted surgery (CAIS) has been suggested to reduce the invasiveness of procedures. Several clinical studies of the surgical team have demonstrated the extreme reliability of these techniques in dental implant placement [22,23].

### 2.3. SINIMAGE Report Outcomes

This randomized controlled trial (RCT) was based on 60 patients: 30 patients were treated using sinus lifting procedures through bone graft (bone graft surgery, BGS), and 30 patients were treated with computer-aided implant surgery (CAIS). The diagram of the study was as reported in Figure 1. A case report form (CRF) was created for each group to allow for a unified treatment approach for patients. This CRF included all details of the treatment as well as answers to questions about the treatment.

#### 2.3.1. Patient-Reported Outcomes after Surgical Intervention

##### Number of Consultations

Patient visits were collected through a CRF of patient’s treatment from radiological study of patients to implantation intervention.

##### Loss of Implants

Loss of one of the implants after one week of implant placement was recorded during the follow-up visit the week after intervention within the treatment CRF.

##### Recommendation of Patients

Recommendation of patients was recorded during the follow-up visit the week after intervention within the treatment CRF by asking the patient whether he would recommend this treatment.

##### Intervention Duration

CRF registered the hour and minute at the start of the implant placement intervention and the hour and minute at which the implant placement intervention was completed.

##### Pain of Implant Placement Intervention

Pain was evaluated in a verbal rating scale (VRS), where adjectives are used to describe different levels of pain [24]. The intensity of pain according to the different terms proposed that they were presented verbally to the patient and the patient verbally assessed the intensity of his pain. The terms were defined in four groups: 1—null, 2—moderate, 3—significant, and 4—severe.

##### Pain on the Intervention Day

The patient’s CRF recorded the pain of treatment in the intervention day.

##### Pain during the Week after the Implant Placement Intervention

The patient’s CRF recorded the pain of treatment the week after implant placement intervention.

##### Treatment Difficulty

Evaluation of treatment difficulty for the study groups evaluates the patient’s description of the difficulty of treatment using four categories within a CRF, namely, “very difficult”, “difficult” for complicated treatment classification and “not difficult”, and “no opinion” for uncomplicated treatment classification.

##### Complications

Some dental implant complications were defined in the study protocol and were recorded in the treatment’s CRF as follows: (1) patients did not have any complications concerning implant placement; (2) non-osseointegrated implants (no new bone is laid down directly on the implant surface); (3) implant unusable prosthetically (the implant cannot be used as an abutment-implant); (4) pain and discomfort when tightening the abutment-implant; and (5) peri-implant bone loss (peri-implantitis is a pathological condition occurring in tissues around dental implants, characterized by inflammation in the peri-implant connective tissue and progressive loss of supporting bone) [25].

#### 2.3.2. Patient-Reported Outcomes after One Year

A clinical examination was performed for all patients after one year of loading implant prostheses and the study protocol identified several points:

##### Peri-Implantitis

Implants infected with complex flora were present with peri-implantitis and close to active periodontitis. Revealing an inflammatory lesion of the peri-implantmucositis, and peri-implantitis, this also includes loss of supporting bone [26].

##### Patient Satisfaction

Assessment of patient satisfaction: the patients expressed if the treatment answered their expectation. The answers were on a scale of four levels (not at all, little satisfaction, satisfied, and very satisfied).

##### Evaluation Criteria of the Success (Defined by the Study Protocol):

Pain around one of the implants (pain post a loading prosthetic implants)(1)Stability of implants (determine the status of implant stability after the loading prosthetic implants)(2)Infections signs around one of the implants (infected dental implant is similar to those of gingiva disease: red or puffy gingiva around the implant, throbbing pain or discomfort or exudates pus from the area, dull ache on palpation);(3)Radiography shows around one of the implants (X-rays show a radiolucent area around one of implants and revealed an abnormal image is present);(4)Occurrence of adverse events (patient has an adverse events since the last visit);(5)Radiography shows losing of supporting bone (radiographic evaluation of implants with emphasis toward crestal bone levels and radiography shows an osseous crater);(6)Loss of one of the implants (several things can cause implant bone loss and occurrence of bone loss around implant will lead to implant loss);(7)Plaque accumulation around implants.The study protocol evaluated the plaque accumulation through the clinical examination and within four categories:(i)No plaque detection;(ii)Plaque only recognized by running a probe across cervical margin of the tooth;(iii)Plaque visible to the naked eye;(iv)Abundant plaque.(8)Bleeding on probing (BOP)Clinical examination determine the patient’s periodontal stability status and subgingival bacterial deposits, and the study protocol evaluated BOP through four categories:(i)No bleeding on probing;(ii)Visible bleeding points;(iii)Red line bleeding on the marginal gingiva;(iv)Abundant bleeding.(9)Presence of keratinized gingivalClinical examinations detect the presence of keratinized gingiva and the keratinized gingiva around implants is associated with the health of implant-supporting tissues.

### 2.4. Statistical Analysis

The null hypothesis of no difference in PROMs between the two techniques of implant placement was used with a confidence interval (CI) of 95%. Quantitative data were described by mean and range and SD (standard deviation) and the Mann–Whitney test was used to compare differences between the two groups and to know the *p*-value. They were also described by the median value with the interquartile range (IQR). Qualitative data were described as the effective number with percent value for the variable. The crosstab variables were used for examining the relationship between two categorical variables and the chi-square test of association was used to discover if there was a relationship between two categorical variables.

## 3. Results

Graft surgery: 30 patients were included in the graft surgery group for implant placement after lifting the sinus membrane.

First stage: 29 patients received the bone graft after lifting the Schneiderian membrane; one patient exited the study owing to depression, which did not allow further sinus treatment.

Second stage: 27 patients had placed 75 implants in the second stage; one patient asked to be removed from the study and one patient was removed from the study after his bone graft failed and the implants were placed by CAIS.

Computer-Aided Implant Surgery: 30 patients were included in the computer-aided implant surgery (CAIS) group (3D planning software, and 3D imaging by CBCT, guided surgical template, and computer-aided surgery).

Twenty-nine patients had placed 69 implants. Then, however, one patient dropped out of the study because there was a mistake in his entering the study (owing to a misinterpretation on the scan reading, the patient did not require grafting); flow-chart for SINIMAGE (Figure 2).

The women ratio was fairly equal among the two surgical techniques, 19 women (63.3%) for graft surgery compared with 18 women (60.0%) for CAIS surgery, A chi-square test for independence indicated no significant difference in the proportion of males or females who were treated by both types of surgery (graft, CAIS), *p* = 0.791.

The ages were fairly close among the two types of surgery, that is, graft surgery μ = 56.7; SD = 9.16, range = (35–73) and CAIS surgery μ = 59.5: SD = 8.96, range = (30–69). There was no significant difference in age between the two types of surgery; *p* = 0.809 (Table 1).

Implant placement by CAIS surgery was planned and placed close to the wall of sinus and its shape and size determined the planning and the implant placement of the sinus region.

The mean number of implants placed was three implants for graft surgery and two implants for CAIS surgery and showed an equal value among the two types of surgery for interquartile range (IQR) [2,3]. There was significant difference in the number of implants placed for both types of surgery; the surgery types affect the number of implants placed, *p* = 0.0336, and this was as a result of the size and shape of the sinus maxillary.

### 3.1. Patient-Reported Outcome for Surgical Intervention

#### Number of Consultations

This concerns the visits from radiographic image studies of patients to the day of implant placement. The Mann–Whitney independent test was performed; there was a significant difference in the number of visits for each surgical technique M = 8 consultations, the interquartile range = [5–10] for graft surgery and M = 4 consultations, while the interquartile range = (3–5) for CAIS; *p* = 0.0001 (Table 2).

These results suggest that the type of surgery has an effect on the number of visits and the mean number of visits for graft surgery was higher than for CAIS at the 5%threshold (graft = 8 consultations ± 3 consultations vs. CAIS = 5 consultations ± 4 consultations).

### 3.2. Loss of Implants

The number of patients who lost their implants during the eight days after implant placement was 2 (6.9%) for CAIS, whereas no patients lost their implants during the eight days after implant placement for graft surgery. *p* = 0.492 > 0.05, so we do not reject the null hypothesis (H0), according to which there is no relation between implant loss and the two surgical techniques at the threshold of 5%. The exact Fisher test for independence indicated no significant difference in the implant loss for both surgical techniques; there was no relationship between implant loss and the surgical techniques *F*-tests (*n* = 56); *p* = 0.492.

### 3.3. Recommendation of Patients

The proportion of patients who recommended the treatment was 23 patients (92.0%) for graft surgery versus 100.0% of patients for CAIS.

*p* = 0.210 > 0.05, so we do not reject the null hypothesis (H0) that there is no relationship between advice to place implants and the type of surgery at the threshold of 5%.

The exact Fisher test for independence indicated no significant difference in the recommendation of patients for implant placement for both types of surgery. There wasno relationship between the recommendation of patients and the two surgical techniques, *F*-tests (*n* = 56); *p* = 0.210.

### 3.4. Intervention Duration 

The duration of implant placement procedure was fairly equal for both types of surgery, μ = 74.56 min, range = (30–135), SD = 31.66 for graft surgery and μ = 72.79 min, rang e = (20–150); SD = 31.65 for computer-aided implant surgery. The median duration of the intervention was same in the two groups: 75 min (IQR: 45–90) in the group graft versus 75 min (IQR: 48–90) in the CAIS group; *p* = 0.9869. The Mann–Whitney independent test was performed and there was no significant difference between the duration of implant placement for the two surgical techniques. These results suggest that surgery types do not have an effect on the duration of implant placement intervention (Table 2).

### 3.5. Pain of Implant Placement Intervention

#### 3.5.1. Pain on the Intervention Day 

The VRS results of pain on the implant placement day showed no statistically significant differences and the type of surgery did not have an effect on pain on the implant placement day. The pain during the surgical operation was approximately similar in the two groups at the threshold of 5% (37.9% in the CAIS group vs. 429.6% in the graft group) (*n* = 56; *p* = 0.512) (Table 3).

The proportions of VRS pain on the implant placement day for graft surgery were as follows: 19 patients had null pain (70.4%), 7 patients had moderate pain (25.9%), 1 patient had significant pain (3.7%), and no patients had severe pain.

On the other hand, the proportions of VRS for CAIS were as follows: 18 patients had null pain (62.1%), 9 patients had moderate pain (31.0%), no patients had significant pain, and 2 patients had severe pain (6.9%).

#### 3.5.2. Pain during the Week after the Implant Placement Intervention 

The VRS results of pain during the week after the implant placement intervention showed no statistically significant differences between the types of surgery. The pain during the week was approximately similar in the two groups at the threshold of 5% (34.5% in the CAIS group vs. 48.2% in the graft group) (*n* = 56; *p* = 0.299). The proportions of VRS pain during the week after the implant placement for graft surgery were as follows 14 patients had null pain (51.9%), 9 patients had moderate pain (33.3%), 3 patients had significant pain (11.1%), and 1 patient had severe pain (3.7%). On the other hand, the proportions of VRS for CAIS were as follows: 19 patients had null pain (65.5%), 9 patients had moderate pain (31.0%), 1 patient had significant pain (3.4%), and no patients had severe pain (Table 4).

### 3.6. Treatment Difficulty

Patients described the difficulty of treatment, which was divided into four categories: “very difficult”, “difficult”, “not difficult”, and “no opinion”. There were no patients without an opinion.

Both “very difficult” and “difficult” were considered a difficult treatment. Nine patients considered graft surgery to be difficult treatment (33.3%) and three patients considered CAIS to be a difficult treatment (10.3%). The chi-square test for independence indicated a statistically significant difference between the treatment difficulty and the two surgical techniques (graft, CAIS) and the surgical techniques had an effect on the treatment difficulty, X2 (*n* = 56); *p* = 0.036 (Table 5).

Patients for graft surgery described treatment difficulty as follows: 2 patients described it as being very difficult (7.4%), 7 patients described it as being difficult (25.9%), and 18 patients described it as being not difficult (66.7%).

On the other hand, patients for computer-aided implant surgery described treatment difficulty as follows: no patient described it as being very difficult, 3 patients described it as being difficult (10.3%), and 26 patients described it as being not difficult (69.7%).

### 3.7. Complications 

Implant complications were recorded at the loading prosthesis stage of the CRFs treatment; two patients from CAIS were removed from the study (one patient because of departure abroad and another because of implant rejection) (Figure 2).

Implant complications were determined by lack of implant osseointegration, implants that were unusable prosthetically, pain and discomfort when tightening the abutment-implant, and peri-implantitis around the implant.

No patient had implants that were unusable prosthetically and three patients had implant complications by a proportion of 11.1% in both types of surgery (Table 6).

Implant complication proportions for graft surgery were as follows: 1 patient had no osseointegration (3.7%), 2 patients had peri-implantitis (7.4%), 1 patient had pain tightening the abutment (3.7%), and 1 patient had pain tightening the abutment and peri-implantitis. Moreover, implant complication proportions for CAIS were three patients who had no osseointegration (11.1%).

The *p*-value associated with Fisher’s exact test was *p* = 1.00 > 0.05, so we do not reject the null hypothesis (H0) that there is no relation between implant complication and the type of surgery at the threshold of 5%.

An exact Fisher test for independence indicated no significant difference in the proportion of implant complication for both types of surgery, *F*-tests (*n* = 54); *p* = 1.00.

The ratio of patients who have complication was 11.1%, nearly equal among both types of surgery.

An exact Fisher test for independence indicated no significant difference in the proportion of implant complication for both types of surgery, *F*-tests (*n* = 54); *p* = 1.00.

### 3.8. Clinical Examination for the First Year

Two patients from the graft group were removed from the study (one patient decided to use the current implants with maxillary over denture retained by the Locator^®^ system and the definitive prosthesis did not load for the other patient) (Figure 2).

#### 3.8.1. Peri-Implantitis 

Peri-implantitis was nearly equal between the two types of surgery, that is, 1 patient (4.0%) for graft surgical versus no patient for CAIS (Table 7).

Fisher’s exact test for independence showed no significant difference in the proportion of peri-implantitis for both types of surgery (graft, CAIS) (*n* = 54; *p* = 0.481 > 0.05).

#### 3.8.2. Patient Satisfaction 

Satisfaction was recorded by the verbal rating scale (VRS) (Table 7):

Very satisfied was nearly equal between the two types of surgery, with 18 patients (72.0%) for graft surgical compared with 21 patients (77.78%) for CAIS; satisfied was 7 patients (28.0%) for graft surgical versus 4 patients (14.81%) for CAIS; and little satisfied was 2 patients (7.41%) for CAIS versus no patients for graft surgery.

No patient gave a “not satisfied”.

Patients were also requested to answer to the question: would you recommend the treatment

#### 3.8.3. Evaluation Criteria of the Success

Clinical examination has identified several criteria for successful treatment (defined by the study protocol) (Table 8):(1)Pain around one of the implants: The percentage of patients who had pain around one of the implants was one patient (4.0%) for graft surgery versus no patients (0.0%) for CAIS. Fisher’s exact test for independence indicated no significant difference in the proportion of pain around one of the implants for both surgeries (*n* = 52, *p* = 0.481 > 0.05).(2)Stability of implants: Stability of the implants was equal between the two groups (graft, CAIS). The implants for all patients were stable (25 patients, 100% for graft surgery and 27 patients, 100% for CAIS).(3)Signs of infections around one of the implants: The presence of signs of infections around one of implants was checked for all patients through the clinical examinations. The percentage of patients who had presence of signs of infections was two patients (7.41%) for CAIS versus no patients (0.0%) for graft surgery. Fisher’s exact test for independence indicated no significant difference in the proportion of presence of signs of infections around one of the implants for both surgeries (*n* = 52; *p* = 0.491 > 0.05).Abnormal radiographic imaging of implants: The percentage of patients who had abnormal image was two patients (8.0%) for graft surgery versus no patients (0.0%) for CAIS. Fisher’s exact test for independence indicated no significant difference in the proportion of X-ray disclosures radiolucent areas for both surgeries (*n* = 52; *p* = 0.226 > 0.05).(4)Occurrence of adverse events: The number of adverse events since the last visit was nearly equal between the two groups (one patient, 4.0% for graft surgery and one patient, 3.7% for CAIS). Fisher’s exact test for independence indicated there was no relationship between the occurrence of adverse events and the technique of surgery (*n* = 52; *p* = 1.00 > 0.05).(5)Radiology evaluation of craterization: The proportion of bone craterization in graft surgery is higher than in CAIS surgery (five patients, 20.0% for graft surgery versus no patients, 0.0% for CAIS). Fisher’s exact test for independence indicated significant difference in the proportion of bone craterization for both surgeries this means that there was a relation between the disclosure of an osseous crater and the technique of surgery at the threshold of 5% (*n* = 52; *p* = 0.020 < 0.05).(6)Loss of one of the implants: One patient from a graft surgery and one from CAIS suffered implant loss (one patient, 4.0% for graft surgery versus one patient, 3.7% for CAIS). Fisher’s exact test for independence indicated no relationship between the loss of an implant and the type of surgery at the 5% threshold (*n* = 52; *p* = 1.00 > 0.05).(7)Plaque accumulation around implants: One patient has a plate on the circumference of the cervix plus a visible plate of the naked eye in graft surgery and one patient from each group had no information.Plaque accumulation was evaluated within four categories.(i)No plaque detectionPatients who had no plaque detection were nearly equal for the two surgeries (22 patients, 88.0% for graft surgery versus 23 patients, 85.19% for CAIS).Chi-square test for independence indicated that there was no relation between no plaque detection and the technique of surgery at the threshold of 5% (*n* = 52; *p* = 1.00 > 0.05).(ii)Plaque at the cervical marginPlaque only recognized by running a probe across cervical margin of the tooth.The number of patients who had plaque at the cervical margin was higher for CAIS (one patient, 4.0% for graft surgery versus three patients, 11.11% for CAIS). Fisher’s exact test for independence indicated no relation between patients who had plaque at the cervical margin surgeries at the threshold of 5% (*n* = 52; *p* = 0.611 > 0.05).(iii)Plaque visible to the naked eyeThe number of patients who had plaque visible to the naked eye was equal in both surgeries (one patient, 4.0% for graft surgery versus one patient, 3.7% for CAIS). Fisher’s exact test for independence indicated no relationship between patients who had plaque visible and the type of surgery at the 5% threshold (*n* = 52; *p* = 1.00 > 0.05).(iv)Abundant plaquePatients who had abundant plaque were missing in CAIS surgery (one patient, 4.0% for graft surgery versus no patients, 0.0% for CAIS). Fisher’s exact test for independence indicated no relationship between abundant plaque and the technique of surgery at the 5% threshold (*n* = 52; *p* = 0.481 > 0.05).(8)Periodontal probing: In graft surgery, one patient had visible bleeding points added to red line bleeding on the marginal gingiva and another one had visible bleeding points and red line bleeding on the marginal gingiva plus abundant bleeding. The periodontal probing index was done through the clinical examination within four categories:(i)No bleeding on probingPatients who had no bleeding on probing for CAIS were nearly equal among both types of surgery (25 patients, 100.0% for graft surgery versus 24 patients, 88.89% for CAIS surgery).Fisher’s exact test for independence indicated no relation between patients who did not have bleeding on probing and both surgeries at the threshold of 5% (*n* = 52; *p* = 0.236 > 0.05).(ii)Visible bleeding pointsThe number of patients who had visible bleeding points was higher for CAIS surgery than graft surgery (three patients, 11.11% for CAIS surgery versus no patients, 0.0% for graft surgery). Fisher’s exact test for independence indicated no relation between patients who had visible bleeding points and both surgical techniques at the threshold of 5% (*n* = 52; *p* = 0.236 > 0.05).(iii)Red line bleeding on the marginal gingivaNo patients had red line bleeding on the marginal gingiva in both surgeries.(iv)Abundant bleedingNo patients had abundant bleeding in both surgeries.(9)Presence of keratinized gingiva: The number of patients who had keratinized gingiva was higher in CAIS surgery (27 patients, 100.0% for CAIS surgery versus 23 patients, 92.0% for graft surgery). Fisher’s exact test for independence indicated no relation between patients who had keratinized gingiva and both surgical techniques at the threshold of 5% (*n* = 52; *p* = 0.226 > 0.05).

## 4. Discussion

This single center trial was conducted with the aim of comparing two procedures of implant placement. The implant placements were performed according to the protocols followed for the two types of surgery—30 patients by graft surgery, which is considered the ideal treatment for sinus lifting; and 30 patients by CAIS, which uses guided surgery, 3D imaging, and software planning.

The analyses of information on patients’ recommendation, difficulty of surgery, complications, surgical pain, and surgery duration were called reported outcome measures (PROMs); and these reported outcome measures were compared for two procedures of implant placement. These outcome measures were reported under the case report form (CRF) of the treatment groups.

There was no significant difference between the surgical techniques for, implant complication, loss of implant, pain evaluation, and implant placement duration. There was not correlation between the types of surgery and those characteristics.

The surgical time of implant placement was approximately equal in both groups and 75 min was the surgical duration in both groups. The use of a surgical guide did not have a clear effect on the duration of surgery, and this may be owing to the expertise of surgeons.

All patients in the CAIS group would recommend the CAIS treatment, although 2 of them (6.9%) had lost implants, 11 patients (7.93%) had pain on the intervention day, and 10 patients (34.48%) had pain during the week after implant placement intervention. On the other hand, for the graft surgery, no patients had loss of implants, 4 patients (8.0%) would not recommend the graft treatment, 8 patients (29.63%) had pain on the intervention day, and 13 patients (48.15%) had pain during the week after implant placement intervention.

The patients in the graft surgery who had pain after a week increased from 8 patients (had pain on the intervention day) to 13 patients, and this was why patients were not recommended for the graft surgery. Moreover, the number of patients who prescribed treatment as difficult was greater in the graft surgery (9 patients, 33.3%) compared with CAIS (3 patients, 10.3%), and the type of surgery affected this result (*p* = 0.036).

The uses of computer-aided navigation technology can significantly improve the quality and increase safety during operation [27]. The study results showed equality in patients who suffered from implant placement complications. Three patients had complications in each of the two operations. The only reason for the complications was no osseointegration in CAIS; for graft surgery, one patient had no osseointegration, one patient had peri-implantitis, and one patient had pain of tightening the abutment screw and peri-implantitis.

Maxillary sinus grafting is a predictable procedure that has been routinely performed for more than 30 years and has a low rate of postoperative complications observed in the grafted area [28].

Computer-based surgical planning allows surgeons to evaluate bone morphology [29] and the use of the CAD surgical guide reduces complications [30].

However, the surgical techniques influenced the number of patient visits, which was the highest in graft surgery. This was logical because of the difference in the surgical stages between the two techniques. The graft surgery of implant placement consists of two stages. During the first stage of treatment, the sinus lift was performed and then the bone graft was placed. In the second stage, after 3 to 6 months, the implants were loaded [31]. On the contrary, the use of computer-aided implant surgery (CAIS) for implant placement required only one surgical stage.

In a similar fashion, most of the results of clinical examination for the first year were fairly close for the two implant placement surgeries.

Most patients were satisfied with the treatment and very satisfied was the highest expression ratio of the satisfied patients (72.0% for graft surgical and 77.78% for CAIS). Only one patient with peri-implantitis was diagnosed in graft surgery.in addition, the majority of patients, were satisfied; this was evidence of the success and reliability of both treatments.

The clinical examination results of the criteria of the success were recorded according to the CRF of the treatment group and were defined by the study protocol (pain, stability, infections signs, abnormal radiographic imaging of implants, occurrence of adverse events, craterization of bone, loss of one of the implants, plaque accumulation around implants, periodontal probing).

All of the clinical examinations except for one did not show statistically significant differences, which indicates the reliability of both techniques for the placement of dental implants.

All implants were stable in clinical examination, while one patient in both treatments lost one of the implants.

None of the patients in treatment with CAIS suffered pain or radiology transparency around one of the implants, while two patients (7.41%) had signs of infection around one of the implants.

On the other hand, one patient was suffered pain and two patients (8.0%) had radiology transparency around one of the implants, while none of the patients had signs of infection around one of the implants in treatment with graft surgery. Moreover, two patients of the graft surgery group did not have presence of keratinized gingival, versus no patients in CAIS.

Bone craterization was higher in graft surgery (five patients, 20.0%) versus no patients for CAIS; the statistical test indicated a significant difference in the proportion of bone craterization for both surgeries and there was a relation between the bone craterization and the technique of surgery at the threshold of 5% (*p* = 0.020).

CAIS was better than graft surgery for no plaque accumulation. There was little detection of plaque around implants, 88.0% for patients in the graft surgery group compared to 85.19% in the CAISgroup. All of patients had no bleeding on probing in graft surgery versus just visible bleeding points was found in three patients for CAIS with a proportion of 11.11%.

The results of comparison were nearly similar with a simple preference for CAIS as the use of a surgical guide for planning of the implant locations improved the success rate and reduced possible complications from the implant surgery [32].

## 5. Conclusions

Graft surgery, compared with computer-aided implant surgery, increased the number of patient visits as well as the number of patients who considered treatment to be difficult, and there were statistically significant differences between the two surgical treatments. The number of patients suffering pain after a week in graft surgery increased from 8 patients (day-intervention pain) to 13 patients. On the other hand, there was a relation between the peri-implantitis and bone craterization in the clinical examination for the first year, with one patient who had peri-implantitis and bone craterization in graft surgery versus no patients in CAIS. Moreover, there were no statistically significant differences regarding the other patient-reported outcomes for surgical techniques and other outcomes of the clinical examination for the first year for implant placement in the sinus region by either bone graft or CAIS.

## Figures and Tables

**Figure 1 ijerph-17-02990-f001:**
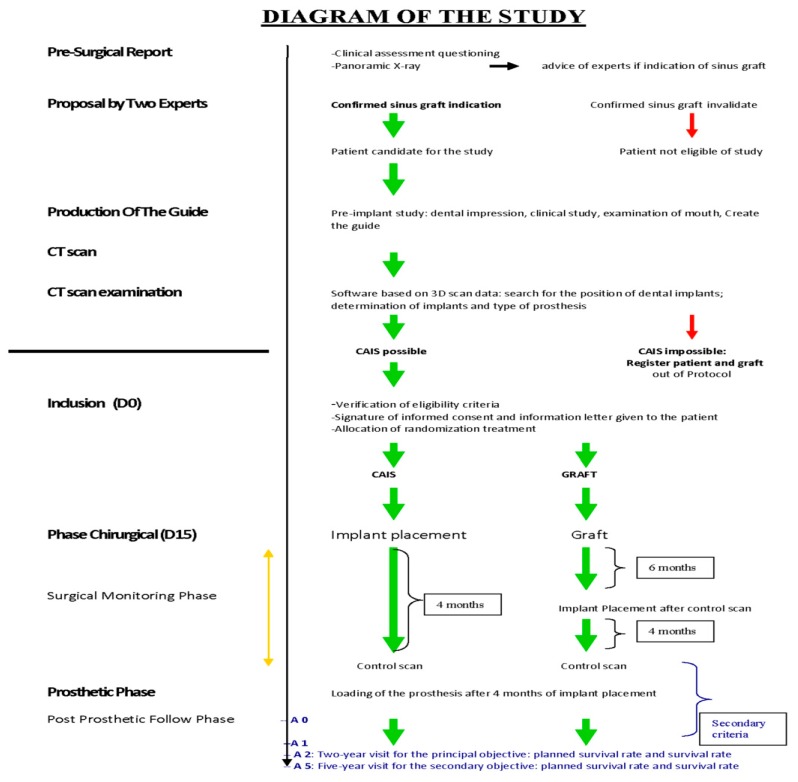
Diagram of the study. CAIS: computer-aided implant surgery. CT: computed tomography; A: year; D: day.

**Figure 2 ijerph-17-02990-f002:**
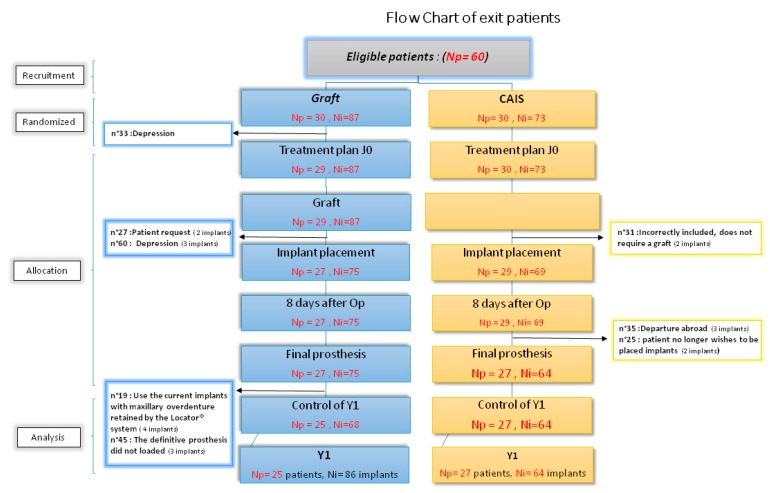
Flow chart of SINIMAGE. Y: year; CAIS: computer assisted implant surgery; SINIMAGE: SinusImage (the study name); Np: number of patients; Ni: number of implant; n°: patient number; Op: surgical opération.

**Table 1 ijerph-17-02990-t001:** Population characteristics. IQR, interquartile range. CAIS, computer-aided implant surgery.

Population Characteristics			
	Graft (*n* = 30)	CAIS (*n* = 30)	*p*-Value
Sex (effective (percentage))	19 women (63.3%)	18 women (60.0%)	0.791
Age (mean (range))	56.7 years (35–73)	59.5 years (30–69)	0.809
Number of implants placed (mean ((IQR))	3 implants (2–3) (*n* = 27)	2 implants (2–3) (*n* = 29)	0.0336

**Table 2 ijerph-17-02990-t002:** Surgery report of interventions.

Surgery Report of Interventions			
	Graft (*n* = 27)	CAIS (*n* = 29)	*p*-Value
Number of consultations (median (IQR))	8 consultations (5–10)	4 consultations (3–5)	0.0001
Duration of implant placement (median (IQR))	75 min (45–90)	75 min (48–90)	0.987
Loss of implants (effective (percentage))	0	2 patients (6.9%)	0.492
Number of patients who recommend the treatment (effective (percentage))	23 patients (92.0%)	29 patients (100.0%)	0.210

**Table 3 ijerph-17-02990-t003:** Pain on the intervention day. VRS, verbal rating scale.

Pain on the Intervention Day				
		Graft (*n* = 27)	CAIS (*n* = 29)	*p*-Value
Intensity of pain VRS	Pain 19 patients (33.93%)	8 patients (29.63%)	11 patients (37.93%)	0.512
	Null 37 patients (66.67%)	19 patients (70.4%)	18 patients (62.1%)	
	Moderate 16 patients (28.51%)	7 patients (25.9%)	9 patients (31.0%)	
	Significant 1 patient (1.79%)	1 patient (3.7%)	0	
	Severe 2 patients (3.57%)	0	2 patients (6.9%)	

**Table 4 ijerph-17-02990-t004:** Pain one week after intervention.

Pain 1 Week Postoperative				
		Graft (*n* = 27)	CAIS (*n* = 29)	*p*-Value
Intensity of pain VRS	Pain 23 patients (41.07%)	13 patients (48.15%)	10 patients (34.48%)	0.299
	Null 33 patients (58.93%)	14 patients (51.9%)	19 patients (65.5%)	
	Moderate 18 patients (32.14%)	9 patients (33.3%)	9 patients (31.0%)	
	Significant 4 patients (7.14%)	3 patients (11.1%)	1 patient (3.4%)	
	Severe 1 patient (1.79%)	1 patient (3.7%)	0	

**Table 5 ijerph-17-02990-t005:** Difficulty of treatment.

Difficulty of Treatment				
(Effective (Percentage))		Graft (*n* = 27)	CAIS (*n* = 29)	*p*-Value
Treatment considered as difficult (Very difficult and Difficult)		9 patients (33.3%)	3 patients (10.3%)	0.036
The evaluation of difficulty of surgical treatment	Very difficult	2 patients (7.4%)	0	
	Difficult	7 patients (25.9%)	3 patients (10.3%)	
	Not difficult	18 patients (66.7%)	26 patients (89.7%)	
	No opinion	0	0	

**Table 6 ijerph-17-02990-t006:** Implant complications.

Implant Complications				
(Effective (Percentage))		Graft (*n* = 27)	CAIS (*n* = 27)	*p*-Value
Implant complications		3 patients (11.1%)	3 patients (11.1%)	1.00
Types of implant complications	No osseointegration	1 patient (3.7%)	3 patients (11.1%)	
	Implant unusable prosthetically	0	0	
	Peri-implantitis	2patients (7.4%)	0.00%	
	Pain tightening the abutment	1 patient (3.7%)	0.00%	

**Table 7 ijerph-17-02990-t007:** The first year of examination.

(Effective (Percentage))		Graft (*n* = 25)	CAIS (*n* = 27)	*p*-Value
Peri-implantitis		1 patient (4.0%)	0	0.0481
Patient satisfaction	Very satisfied	18 patients (72.0%)	21 patients (77.78%)	
	Satisfied	7 patients (28.0%)	4 patients (14.81%)	
	Little satisfied	0	2 patients (7.41%)	

**Table 8 ijerph-17-02990-t008:** Evaluation criteria of the success.

Evaluation Criteria of the Success (Effective (Percentage))				
		Graft (*n* = 25)	CAIS (*n* = 27)	*p*-Value
Pain around one of the implants		1 patient (4.0%)	0	0.481
Stability of implants		25 patients (100%)	27 patients (100%)	
Infectious signs around one of the implants		0	2 patients (7.41%)	0.491
Abnormal radiographic imaging		2 patients (8.0%)	0	0.226
Occurrence of undesirable events since the last visit		1 patient (4.0%)	1 patient (3.7%)	1.00
Radiology evaluation of craterization		5 patients (20.0%)	0	0.020
Loss of one of the implants		1 patient (4.0%)	1 patient (3.7%)	1.00
Plaque accumulation around implants	No plate detection	22 patients (88.0%)	23 patients (85.19%)	1.00
	plaque at the cervical margin	1 patient (4.0%)	3 patients (11.11%)	0.611
	Plate visible to the naked eye	1 patient (4.0%)	1 patient (3.7%)	1.00
	Abundant plaques	1 patient (4.0%)	0	0.481
Periodontal probing	No bleeding on probing	25 patients (100.0%)	24 patients (88.89%)	0.236
	Visible bleeding points	0	3 patients (11.11%)	0.236
	Red line bleeding on the marginal gingiva	0	0	0
	Abundant bleeding	0	0	0
Presence of keratinized gingiva		23 patients (92.0%)	27 patients (100%)	0.226

## Data Availability

The datasets are available from the author upon reasonable request.

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
