# Peer review of "Patient-Reported Outcome Measures (PROMs) for Two Implant Placement Techniques in Sinus Region (Bone Graft versus Computer-Aided Implant Surgery): A Randomized Prospective Trial"

_ijerph, 2020, doi:10.3390/ijerph17092990_

Round 1

Reviewer 1 Report

very interesting work that present the patient-reported outcomes of two different treatments available for implant rehabilitation: sinus graft vs guided implant surgery.

the study is well written but it would benefit for a careful revision of the english form (x example, "plaque" accumulation- and not "plate" accumulation, and some other minor typos) that can be improved and i have some minor comments the authors should address to improve the overall quality.

1. intro/ the intro is too short and does not provide a sufficient background for the study, particularly at the beginning, the authors fail to introduce the world of digital dentistry properly and in sufficient details. i suggest the authors to write at least one paragraph to dedicate to digital dentistry and the description of the different new devices and software, with proper references. this is useful to introduce the concept of guided surgery. you should add this paragraph with some proper references then you should add the new references to your list then you have to renumber all your references in the text and in the reference list accordingly

2. methods/ can you highlight the references in the square parenthesis, in the text, using a different colour (blue)?

3. results/ please check your results in particular the tables and the related legends in order they are more readable in the text

4. discussion/ discussion is ok

5. references/ the list is too short and should be enlarged. ref. 6 is not acceptable. 

6. tables/ figures/ are ok.

Author Response

Reviewer 1

very interesting work that present the patient-reported outcomes of two different treatments available for implant rehabilitation: sinus graft vs guided implant surgery.

THANK YOU VERY MUCH.

the study is well written but it would benefit for a careful revision of the english form (x example, "plaque" accumulation- and not "plate" accumulation, and some other minor typos) that can be improved and i have some minor comments the authors should address to improve the overall quality.

WE HAVE CAREFULLY REVISED OUR ENGLISH FORM.

  1. intro/ the intro is too short and does not provide a sufficient background for the study, particularly at the beginning, the authors fail to introduce the world of digital dentistry properly and in sufficient details. i suggest the authors to write at least one paragraph to dedicate to digital dentistry and the description of the different new devices and software, with proper references. this is useful to introduce the concept of guided surgery. you should add this paragraph with some proper references then you should add the new references to your list then you have to renumber all your references in the text and in the reference list accordingly.

WE HAVE INSERTED A NEW PARAGRAPH FOR THE INTRODUCTION OF THE DIGITAL DENTISTRY. THIS PARAGRAPH HAS 10 NEW REFERENCES AND ACCORDINGLY WE HAVE ADDED THESE NEW REFERENCES TO THE REFERENCE LIST AND WE HAVE RENUMBERED ALL REFERENCES IN THE TEXT TOO.

  1. methods/ can you highlight the references in the square parenthesis, in the text, using a different colour (blue)?

WE HAVE HIGHLIGHTED THE REFERENCES IN THE SQUARE PARENTHESIS USING A DIFFERENT COLOUR (BLUE) IN ADDITION ALL REFERENCES HAVE BEEN RENUMBERED THEREFORE WE HAVE HIGHLIGHTED THEM IN YELLOW.

  1. results/ please check your results in particular the tables and the related legends in order they are more readable in the text

WE HAVE CHECKED THE FORMAT OF OUR RESULTS IN ORDER THEY ARE MORE READABLE.

  1. discussion/ discussion is ok

THANK YOU.

  1. references/ the list is too short and should be enlarged. ref. 6 is not acceptable.

WE HAVE ENLARGED THE NUMBER OF REFERENCES WE HAVE NOW 32 INSTEAD OF 22 IN THE LIST. WE HAVE REPLACED REF. 6 THAT WAS NOT ACCEPTABLE.

  1. tables/ figures/ are ok.

THANK YOU.

Reviewer 2 Report

overall the study is valuable. i suggest the authors to expand their introduction because less than 1 page is not enough. they should consider more studies for example on the cone beam computed tomography that makes guided implant surgery easily accessible to everybody now with lower doses of radiations for the patient. in addition the authors should better organize the result session. i feel not easy to read this session for the reader. the number of references needs also to be extended. the quality of the english language should be checked by a native speaker before resubmission. 

Author Response

Reviewer 2

overall the study is valuable. i suggest the authors to expand their introduction because less than 1 page is not enough. they should consider more studies for example on the cone beam computed tomography that makes guided implant surgery easily accessible to everybody now with lower doses of radiations for the patient.

THANK YOU VERY MUCH. WE HAVE INTRODUCED THE CONCEPT OF CONE BEAM COMPUTED TOMOGRAPHY AND WE HAVE ADDED A PERTINENT REFERENCE ON THAT. THE INTRODUCTION HAS BEEN THEREFORE ENLARGED AS REQUESTED ALSO BY REVIEWER N° 1.

in addition the authors should better organize the result session. i feel not easy to read this session for the reader.

THE RESULTS SESSION HAS BEEN ORGANIZED IN ORDER TO BE MORE READABLE.

the number of references needs also to be extended.

THE REFERENCES ARE NOW 32 INSTEAD OF 22.

the quality of the english language should be checked by a native speaker before resubmission.

THE ENGLISH LANGUAGE HAS BEEN CAREFULLY REVISED.

Reviewer 3 Report

EXCELLENT ARTICLE I THINK IT WOULD BE ADVISABLE TO ADD THE ECONOMICAL IMPACT OF THE TWO DIFFERENT THERAPIES IN THE NEXT RESEARCH PAPER OF YOURS. WHY YOU DID NOT ADD THE COST OF THE TWO THERAPIES AND MADE A COMPARISON OF THEM? THE ENGLISH FORM SHOULD BE REVISED TOO. 

Author Response

EXCELLENT ARTICLE I THINK IT WOULD BE ADVISABLE TO ADD THE ECONOMICAL IMPACT OF THE TWO DIFFERENT THERAPIES IN THE NEXT RESEARCH PAPER OF YOURS. WHY YOU DID NOT ADD THE COST OF THE TWO THERAPIES AND MADE A COMPARISON OF THEM?

THE ECONOMIC ASPECTS WILL BE CONSIDERED IN DETAILS IN OUR NEXT RESEARCH.

THE ENGLISH FORM SHOULD BE REVISED TOO.

WE HAVE CAREFULLY REVISED OUR ENGLISH FORM.